# A Review of Early Fault Diagnosis Approaches and Their Applications in Rotating Machinery

**DOI:** 10.3390/e21040409

**Published:** 2019-04-17

**Authors:** Yu Wei, Yuqing Li, Minqiang Xu, Wenhu Huang

**Affiliations:** Department of Astronautical Science and Mechanics, Harbin Institute of Technology (HIT), Harbin 150001, China

**Keywords:** early fault diagnosis, rotating machinery, signal processing, feature extraction

## Abstract

Rotating machinery is widely applied in various types of industrial applications. As a promising field for reliability of modern industrial systems, early fault diagnosis (EFD) techniques have attracted increasing attention from both academia and industry. EFD is critical to provide appropriate information for taking necessary maintenance actions and thereby prevent severe failures and reduce financial losses. A massive amounts of research work has been conducted in last two decades to develop EFD techniques. This paper reviews and summarizes the research works on EFD of gears, rotors, and bearings. The main purpose of this paper is to serve as a guidemap for researchers in the field of early fault diagnosis. After a brief introduction of early fault diagnosis techniques, the applications of EFD of rotating machine are reviewed in two aspects: fault frequency-based methods and artificial intelligence-based methods. Finally, a summary and some new research prospects are discussed.

## 1. Introduction

Rotating machinery is the most commonly used type of machine in modern industry, civilian and military applications, such as compressors, steam turbines, automobiles, industrial fans, and aircraft engines [1,2,3,4,5,6,7]. Due to the high service load, harsh operating conditions or inevitable fatigue, faults may develop in rotating machinery [8,9,10]. If the fault cannot be diagnosed in a timely way, it may cause a shutdown of the whole system and even catastrophic failure. Therefore, it is significant to detect faults early and assess the fault level as early as possible to avoid catastrophic accidents and ensure the safe operation of the machinery [11,12,13,14,15,16]. 

Many researchers are committed to developing early fault diagnosis (EFD) techniques to monitor the health conditions of rotating machinery [17,18,19,20,21]. At present, many EFD techniques have been successfully applied in the modern industries, such as the vibration-based EFD method [22], current-based EFD method [23], acoustic emission-based EFD method [24], sound-based EFD method [25], torque-based EFD method [26], and rotating encoder-based EFD method [27], etc. Among these sensor signals, vibration-based diagnostic method is the most commonly researched one because vibration signals can directly represent the dynamic behavior of rotating machinery [28,29,30]. However, the early fault signal is often too low due to the fact that small localized damage may generate small periodical impulses. Meanwhile, the measured vibration signals at the early stage are interfered by strong noises [31]. For example, Figure 1 shows a real whole life bearing run-to-failure process. As seen in the figure, there are three stages in the whole life including the normal operation stage (I), early fault stage (II), and failure stage (III).

Two vibration signals taken from the severe fault stage (phase III) and early fault stage (phase II) are shown in Figure 2a and Figure 2c, respectively. Meanwhile, their corresponding Hilbert transform (HT) spectra are illustrated in Figure 2b and Figure 2d, respectively. It can be found that the serve fault vibration signal has obvious periodical impulses (as shown in Figure 2a) and the fault frequency and its harmonics can be clearly detected in the frequency domain (as shown in Figure 2b). However, the sign of the periodic impulse is totally marked by the noise in the early fault vibration signal (as shown in Figure 2c). Meanwhile, its corresponding fault frequency, which is totally embedded in the background noises, cannot be detected (as shown in Figure 2d). Therefore, the measured vibration signals of weak fault damage are often interfered by strong background noise and it is difficult to detect the localized damage at the early stage.

To diagnose early faults as soon as possible, the feature extraction of vibration signals is very important in real engineering applications. Recently, the advanced signal processing-based weak feature extraction method has been becoming a hot research topic. However, the measured vibration signals of rotating machinery often present nonlinear and non-stationary characteristics. Moreover, the vibration signals contain strong background noises during the early fault stage, making it difficult to extract fault information. In order to effectively analyze the non-stationary vibration signals, massive research efforts have been made in last two decades to develop various signal processing technologies, including wavelet transformation (WT), empirical mode decomposition (EMD), ensemble empirical mode decomposition (EEMD), local mean decomposition (LMD), empirical wavelet transform (EWT), variational mode decomposition (VMD), stochastic resonance, sparse decomposition, etc. In addition, artificial intelligence-based methods are also attracting increasing attention. This paper intends to review the existing EFD-based methods and their applications in rotating machinery. For each EFD-based technique, we will review the applications to gears, rotors, bearings and others, respectively.

This paper was written with the objective to serve as a guidemap for those who study EFD of rotating machinery. There are three reasons for such a review. First, a massive research effort has been made in last two decades to develop EFD techniques However, a comprehensive review on EFD techniques does not exist. Therefore, it is necessary to provide a thorough review on EFD methods and their applications in fault diagnosis of rotating machinery in this paper. The literature review will help other researchers, especially researchers new to the topic, to properly select appropriate EFD methods to diagnose early faults. Second, a full literature review is conducted for gears, bearings, rotors and other rotating machinery, respectively. Researchers who work on EFD of rotating machinery, can thus easily determine the research state-of-art in the use of advanced EFD methods. Our work will help them find proper methods for their specific applications and save them time. Third, although many techniques have been developed, there are still some issues that should be studied in-depth. The insights and experiences shared in this paper will benefit other researchers to further develop new EFD methods.

The organization of the rest of this paper is as follows: Section 2 reviews early fault diagnosis approaches. Section 3 gives the applications of fault frequency detection (FFD) in early fault diagnosis of rotating machinery. Section 4 describes the applications of artificial intelligence (AI) in early fault diagnosis of rotating machinery. The conclusions are drawn in Section 5.

## 2. Review of Early Fault Diagnosis Approaches

The EFD of rotating machinery has attracted substantial attention from both academia and industry. According to the difference of the implementation way for EFD of rotating machinery, we can divide it into two main categories: FFD methods and AI methods. We will briefly introduce the principles of the most commonly used FFD methods, including WT, EMD, EEMD, LMD, EWT, VMD, and sparse decomposition. Then, a framework of AI methods is introduced and the most commonly used classifiers are described in detail.

### 2.1. FFD-Based Early Fault Diagnosis

The FFD-based early fault diagnosis system often consists of three steps, including data acquisition, signal denoising and demodulation analysis. Figure 3 shows the flowchart of the FFD-based early fault diagnosis. Among these steps, signal denosing plays a vitally important role in extracting the weak fault characteristics. Many researchers are committed to developing denosing approaches to complete the early fault diagnosis, such as WT, adaptive decomposition methods, and sparse decomposition. 

#### 2.1.1. Adaptive Decomposition Methods 

Unlike traditional signal processing methods, the adaptive decomposition methods can automatically decompose a signal without skilled human intervention. Meanwhile, the adaptive decomposition methods do not require much prior knowledge of the signals. They are effective in transient representation of signals, which can be used to highlight the local characteristics of a signal caused by faults. Until now, several adaptive mode decomposition methods have been developed, such as empirical mode composition (EMD), ensemble empirical mode composition (EEMD), and local mean decomposition (LMD), empirical wavelet transform (EWT) [32], and variational mode decomposition (VMD) [33]. 

##### Empirical Mode Composition

EMD, proposed by Huang et al [34], can adaptively decompose a complicated multi-component signal into a series of intrinsic mode functions (IMFs), whose instantaneous frequencies have physical meaning. To ensure each IMF is a mono-component, two conditions are given as follows: (I) In the whole time span, the number of extrema and the number of zero crossings must either equal or differ at most by one; (II) At any time, the instantaneous mean of the upper and lower envelopes is zero. A complicated signal x(t) can be reconstructed using IMFs through the EMD decomposition process as expressed in Equation (1):(1)$$x(t)=\sum_{i=1}^{n}IMF_{i}(t)+u_{n}(t)$$where $u_{n}(t)$ is the residual.

EMD can self-adaptively decompose a complicated signal into a series of IMFs, which shows remarkable effectiveness in the rotating machinery fault diagnosis. However, several problems exist in the EMD method, such as boundary effects [35], mode mixing [36] and over- and undershoot problems [37]. In order to overcome these drawbacks of EMD, ensemble empirical mode decomposition (EEMD) is proposed with the help of white noises [38]. The details of EEMD will be given in the following subsection. 

##### Ensemble Empirical Mode Composition

EEMD, proposed by Huang, is inspired by the dyadic filter bank nature of the EMD when applied to white noise [38]. White noise of finite amplitude is added to the signal to provide a uniform reference frame in the time–frequency space, perturb the signal in the neighborhood of its true solutions, and thereby force the ensemble to exhaust all possible solutions in the EMD sifting process, and enable the signal components of different scales to collate in proper IMFs. In the ensemble mean of sufficient trials, the noise will be averaged out since it is different in separate trials. Note that two parameters should be set before using EEMD method, including noise amplitude and running trails. Here, it is suggested to set amplitude as 0.2 × *std* (signal), and the number of trials should be set to a few hundred [38]. 

##### Local Mean Decomposition

LMD is another widely used adaptive decomposition method, which was proposed by Smith in 2005 [39]. LMD can decompose a complicated signal into a series of product functions (PFs) and each PF is in essence the product of an envelope signal *a*(*t*) and a frequency modulated signal *s*(*t*). The LMD of a signal is accomplished by progressively smoothing the signal using moving averaging. It is a self-adaptive decomposition method. Compared with EMD and EEMD, LMD has one main advantage that LMD can directly calculate the instantaneous frequency (IF) of each PF without using the Hilbert transform (HT). With the help of LMD, a signal *x*(*t*) can also be reconstructed using Equation (2):(2)$$x(t)=\sum_{p=1}^{k}PF_{k}(t)+u_{k}(t)$$where *u_k_*(*t*) is the residual signal and *k* is the number of PFs.

Although LMD shows excellent performance in demodulating amplitude and frequency modulated signals, it still suffers from some drawbacks, including end effects, mode mixing, and difficulty in the determination of iteration stop criterion and sliding step size.

##### Empirical Wavelet Transform

Empirical wavelet transform (EWT) is a novel adaptive decomposition method proposed by Gilles [40] in 2013. EWT not only has the merits of wavelet transform and EMD but also overcomes the drawbacks of the wavelet and EMD. The EWT algorithm adopts adaptive wavelets, which are able to extract the AM-FM components of a signal. The key idea of EWT is that such constituent amplitude-modulated and frequency-modulated (AM-FM) components have a compact supporting Fourier spectrum. 

The signal can be reconstructed by inverse empirical wavelet transform as follows:(3)$$x(t)=W(0,t)*\phi_{1}(t)+\sum_{n=1}^{N}W(n,t)*\psi_{n}(t)$$where ∗ denotes the convolution operator. The empirical modes are expressed in Equations (4) and (5):(4)$$x_{0}(t)=W(0,t)*\phi_{1}(t)$$
(5)$$x_{k}(t)=W(k,t)*\psi_{k}(t)$$

The detailed coefficients of the inner products with the empirical wavelets and approximation coefficients by the inner product with the scaling function respectively are written in Equations (6) and (7):(6)$$W(n,t)=\int x(\tau )\psi_{n}(\tau -t)d\tau$$
(7)$$W(0,t)=\int x(\tau )\phi_{1}(\tau -t)d\tau$$

Like WT, EWT separates the empirical mode from low frequency to high frequency, however, the bandwidth is not dyadic.

##### Variational Mode Decomposition

Recently, Dragomiretskiy et al. [33] proposed a novel variational method, called variational mode decomposition (VMD), to decompose a multi-component signal into an ensemble of band-limited intrinsic mode functions (IMFs). It was demonstrated that the VMD performed well in tone detection, tone separation, and noise removal. Therefore, much attention has been focused on VMD applications. VMD is a non-recursive decomposition method, and it extracts the constituent AM-FM components of a complicated multi-component signal adaptively and concurrently. It defines IMFs as explicit AM-FM models, and relates the parameters of AM-FM models to the bandwidth of IMFs. According to the narrow-band property of IMFs, the AM-FM parameters can be found by minimizing the bandwidth, thus obtaining IMFs. This algorithm has good merits over other available mode decomposition methods, such as theoretical rationale and robustness to noise and sampling.

In essence, IMFs are AM-FM signals, and they have a limited bandwidth. VMD decomposes a signal *x*(*t*) into an ensemble of IMFs *c_k_*(*t*) that are band-limited about their respective center frequency *w_k_*, while reconstructing the signal optimally. It iteratively updates each IMF *c_k_*(*t*) in the frequency domain, and then estimates the center frequency *w_k_* as the center of gravity of the IMF power spectrum.

#### 2.1.2. Wavelet Transform

The wavelet transform (WT), which employs the wavelet basis function, was proposed by Morlet in 1974 [41]. WT uses the basis function to match a specific fault symptom. Meanwhile, WT is a kind of inner product transform that analyzes the non-stationary contents of the signal through a pre-determined wavelet basis. WT is suitable to analyze the transient signal from the measured vibration signal [42,43,44,45]. WT is categorized as continuous wavelet transform (CWT), discrete wavelet transform (DWT) and wavelet packet transform (WPT). The detail decerebrations are provided as follows.

The CWT of any signal *x*(*t*) can be computed by integrating the signal with a family of wavelet. CWT can be defined in Equation (8):(8)$$CWT_{x}(\tau ,a)=\frac{1}{\sqrt{a}}\int_{-\infty }^{+\infty }x(t)\varphi (\frac{t-\tau }{a})dt$$where wavelet φ[(t−τ)/a] is derived by dilating and translating the wavelet basis φ(t), *a* is the scale parameter (a>0), and *t* is the time shift.

DWT is the discretization of the scale and translation variables in the CWT. DWT can be described as follows:(9)$$DWT_{x}(r,c)=\frac{1}{\sqrt{2^{r}}}\int x(t)\varphi (\frac{t-c2^{r}}{2^{r}})dt$$where 2*^r^* is the scale parameter, and *c*2*^r^* is the shift parameter. WPT is a substitute for CWT. DWT decomposes one signal in the low region, while WPT decomposes one signal in the high region.

#### 2.1.3. Sparse Decomposition

As a new branch of the signal processing methods, sparse representation has received considerable attention, and has been widely used in the field of EFD of rotating machinery [46,47,48,49,50,51,52].

The main goal of sparse representation is to seek a signal representation that is as concise as possible. In the mathematical sense, a sparse signal representation is desired, i.e., to represent a signal accurately via a few atoms, so the number of nonzero representation coefficients is as small as possible. It is essentially an optimization problem, in which an objective function is denoted by the norm of vectors. Here, the definition of the sparsest or nearly sparsest representation over a dictionary is called sparse representation. Its mathematical model can be formulated as:(10)$$\underset{B,\alpha_{i}}{\min}\sum_{i=1}^{m}{\left|x_{i}-B\alpha_{i}\right|}^{2}+\lambda \sum_{i=1}^{m}\left|\alpha_{i}\right|$$where $x_{i}$ denotes the *i*-th sample, *B* denotes the dictionary matrix, and $\alpha_{i}$ represents the sparse representation of $x_{i}$ and λ represents a positive parameter.

### 2.2. AI-Based Early Fault Diagnosis

As an emerging field in industrial applications and an effective solution for fault recognition, AI-based early fault diagnosis method has been receiving increasing attention from academia and industry [3]. It is known that the abovemenioned FFD-based early fault diagnosis methods require rich expertise of the investigators, which makes them inappropriate for common users. The AI-based early fault diagnosis methods can overcome this limitation effectively by using machine learning techniques.

The AI-based early fault diagnosis methods often consist of data acquisition, fault feature extraction and fault pattern identification steps, as shown in Figure 4. The key step is to extract the fault features for further classification. In most AI methods, feature selection before the fault identification is also indispensable. With the help of feature extraction, features with more important information will be selected to construct the new feature set, aiming to improve the final classification accuracy. In AI-based early fault diagnosis schemes, the most commonly used classifiers are K nearest neighbor (KNN), neural network (NN), and support vector machine (SVM).

#### 2.2.1. KNN

KNN is one of the simplest data mining classification methods. In KNN classification methods, each training sample is described as an S-dimensional space according to the value of each of its S features [53]. The testing sample is then represented in the same space, and its K nearest neighbors can be obtained. The class of each of K nearest neighbors is counted, and the class with the largest number of “votes’’ is chosen as the classification of the testing sample. The K nearest neighbors are usually determined by computing the Euclidean distance between the testing sample and each of the training samples [54]. The Euclidean distance between the testing sample *Test_s_* and the *m*-th training sample *Train_m,s_* is defined in Equation (11): (11)$$D={\left[\sum_{s=1}^{S}{\left(Test_{s}-Train_{m,s}\right)}^{2}\right]}^{1/2}\;s=1,2,\cdots ,S,\;m=1,2,\cdots ,M$$where *S* and *M* represent the number of features and training samples, respectively

In order to show the KNN algorithm schematic clearly, a diagram of the KNN method is illustrated in Figure 5. The simplicity of the KNN method makes it easy and widely applicable in fault classification of rotating machinery.

#### 2.2.2. SVM

SVM, proposed by Vapnik, is one of the most widely used and effective classification algorithms. SVM is a machine leaning algorithm based on statistical learning theory (SLT). The initial idea of SVM is to use a linear separating hyperplane to divide the training samples into two classes. Generally, there are two kinds of suitable methods to complete this target [5]. The first one is to find the optimal decision hyperplane which halves the two closest samples into two convex hulls. Figure 6 shows an example of maximum margin. The second one is to discover the optimal decision hyperplane making the margin between two parallel supporting planes. Both these two methods can produce the optimal decision hyperplane and the support vectors. In practical applications, the problems based on multiple classes instead of two classes are needed to solve. The widely used methods of SVM are such as: one against (OAA) all proposed by Vapnik [55], one against one (OAO) proposed by Knerr [56], decision directed acyclic graph proposed by Platt (DDAG) [57] and binary tree (BT) proposed by Vural and Dy [58]. 

#### 2.2.3. Neural network

Neural network is an operational model consisted of a large number of nodes or neurons, connected to each other. The BP neural network proposed by Rumelhart and McClelland in 1986 is a widely used neural network. A BP neural network is trained by an error propagation algorithm, and is a kind of multilayer feed-forward network.

The BP neural network consists of an input layer, hidden layer and output layer. The structure is shown in Figure 7. The input signals are moved from the input layer nodes to hidden layer nodes, and then to the output nodes. The input layer nodes are equal to the number of fault feature vectors $X=(x_{1},x_{2},\cdots ,x_{n})$, the number of the output layer nodes is equal to the number of failure mode vectors $Y=(y_{1},y_{2},\cdots ,y_{n})$. The number of hidden layers is usually a layer. Through the training, the various states of the training sample can be stored in the network in the form of weight values. The weights and threshold value of the network are constantly adjusted by back propagation, and they help make the network error sum of squares minimum [59]. 

The advantage of the BP neural network is that the output of each layer node only influences the output of the next layer node, and as long as there enough of the hidden layer and the hidden nodes, a BP neural network can approach arbitrary nonlinear hint relations, and it has great generalization ability. 

## 3. Applications of FFD in Early Fault Diagnosis of Rotating Machinery

### 3.1. Adaptive Decomposition Methods

#### 3.1.1. EMD

EMD can self-adaptively decompose a complicated signal into a series of IMFs, which represent the natural oscillations embedded in the vibration signal. Analyzing the IMFs, the health state of rotating machinery can be reflected. Dybała et al. [60] used EMD to decompose the signal into IMFs, then spectral analysis was used to get the IMFs for bearing damage detection at an earlier stage. Zhu et al. [61] applied correlation coefficients to select sensitive IMFs. Then envelope analysis was used to demodulate the selected sensitive IMFs for the EFD of rolling bearings. Dybała et al. [62] applied statistical parameters of IMFs for early gearbox diagnostics. Li et al. [63] utilized bandwidth EMD to decompose the signal into IMFs, then adaptive multiscale morphological analysis was used to demodulate the IMFs for EFD of rolling bearings. Zhao et al. [64] used approximate entropy and EMD to quantitative the spall-like fault of the rolling element bearing in early stage. Lv et al. [65] extended the EMD into MEMD to simultaneously analyze the multivariate signal during the period of early failure. Parey et al. [66] applied EMD to decompose signal into IMFs, then variable cosine window was used for IMFs to minimize boundary distortion problem and statistical parameters of IMFs were used for gearbox diagnosis. Above mentioned method has been summarized in Table 1.

#### 3.1.2. EEMD

EEMD is an improved EMD method designed to overcome the problem of mode mixing. The EEMD-based EFD is similar as in the EMD-based method. Chen et al. [67] used EEMD to decompose signal into IMFs, then Hilbert square demodulation was used to demodulate the selected IMFs for wind turbine gearbox fault diagnosis. Wang et al. [68] utilized a tunable Q-factor wavelet transform to select IMFs, then envelope demodulation was used to detect rolling bearing early weak faults. Žvokelj et al. [69] used ICA to select IMFs, and envelope analysis was used to detect and locate the early-stage rolling–sliding contact fatigue failure of bearings. Chen et al. [70] put the IMFs into adaptive stochastic resonance system for early feature extraction of planetary gears.

Several improved EEMD methods have been proposed to enhance the performance of EEMD in EFD of rotating machinery. Guo et al. [71] proposed an enhanced EEMD, which used the similarity criterion to generate the monocomponent for accurate fault diagnosis of rolling bearings. References [72] and [73] introduced a noise-improved method called complementary EEMD (CEEMD) to detect faults at the early stage of degradation. In this method, a cross-correlation coefficient was used to reduce noise and select effective IMFs. Tabrizi et al. [74] proposed an adaptive method to select the optimal amplitude and ensemble trial number of EEMD for roller bearing damage detection. Jiang et al. [75] used multiwavelet packet as the pre-filter to improve EEMD decomposition results, then the sensitive IMFs were used for multi-fault diagnosis identification of rotating machinery. The abovementioned method are summarized in Table 2.

#### 3.1.3. LMD

LMD decomposes a signal into a series of product functions (PFs). Each PF is the product of an envelope signal and a frequency modulated signal, which can be used to detect early faults. Liu et al. [76] introduced LMD to decompose a signal and the instantaneous frequency was calculated from the frequency modulated signal for fault diagnosis of gearbox in wind turbines. Feng et al. [77] utilized LMD to decompose a signal and the amplitude and frequency demodulation method was applied for fault diagnosis of planetary gearboxes. Wang et al. [78] used LMD and instantaneous time-frequency spectrum for the early detection of gear local faults in a finishing rolling mill. Li et al. [79] utilized differential rational spline-based LMD to decompose the signal into a series of PFs, and then the selected main PF components were used to detect early faults in gears and rolling bearings. The abovementioned methods have been summarized in Table 3.

#### 3.1.4. EWT

The EWT algorithm adopts adaptive wavelets to extract the AM-FM components of a signal. Zhang et al. [80] used EWT to decompose the signal into a set of sub-components. Then, the sensitive component was further input into a bistable stochastic resonance system to enhance the fault features. Finally, the weak fault features were identified from the FFT spectrum of the bistable stochastic resonance output. Chen et al. [81] utilized wavelet spatial neighboring coefficient denoising with a data-driven threshold to increase the SNR. Then EWT was used to extract inherent modulation information by decomposing signals into mono-components under an orthogonal basis. Finally, this method had been applied to identify weak faults and compound fault diagnosis. Boualem et al. [82] proposed a EWT methid which applied an appropriate wavelet filter bank with Hilbert transform in the early detection and condition monitoring of tooth crack fault signals. Lu et al. [83] utilized the kurtogram to locate the fault frequency band and filter out the system noise. Then, the preprocessed signal was filtered using the EWT. The l(q)-regularized sparse regression was implemented to obtain a sparse solution of the defect signal in the frequency domain. These method are summarized in Table 4.

#### 3.1.5. VMD

VMD decomposes a multi-component signal into an ensemble of band-limited intrinsic mode functions (IMFs). These IMFs can then be used to diagnose early rotating machinery faults. Ma et al. [84] used VMD to decompose the signal into IMFs, and the number of IMFs was determined by adaptive scale space spectrum segmentation. Finally, the Teager energy operator of the effective IMF components was applied to realize the early fault identification. Li et al. [85] used improved autoregressive-minimum entropy deconvolution and VMD to extract the periodic transient impulses of single-fault or multiple-faults in rotating machinery. Yang et al. [86] developed an early online chatter identification method for milling processes. In this method, an optimized VMD was used to decompose the signal, and the sub-components containing chatter information were extracted using a simulated annealing algorithm. Finally, the approximate entropy and the sample entropy were used to detect five types of operating conditions. Guo et al. [87] proposed a parameter optimization algorithm to select the crucial VMD parameters. Then, the weak features were extracted by using the VMD with the optimized parameters. In [88], rescaling subsampling compression is used to preprocess the signal, and analytical mode decomposition was used to enhance and select signal components. Then the processed signal was decomposed into IMF by VMD to analyze weak multi-frequency signals. In [89], EMD was used to decompose the signal into IMFs selected by relatedness and kurtosis. Then VMD was used to decompose the reconstructed signal using the selected IMFs. Finally, envelope analysis was used for incipient fault diagnosis of rolling bearing. The methods mentioned above method have been summarized in Table 5.

#### 3.1.6. Other Adaptive Methods

Elasha et al. [90] utilized two adaptive filters least mean squares (LMS) and fast block LMS (FBLMS)) to identify early bearing defects in gearboxes. Zhao et al. [91] proposed a reweighted SVD strategy to denoise the signals and enhance weak features for fault diagnosis of rotating machinery. Ibrahim et al. [92] used a least mean squares algorithm to reduce the noise, and then the meshing frequency sidebands was found to successfully identify gear faults. Mei et al. [93] used linear multi-scale segmentation to divide the signal into segments with nearly linear frequency, then the multi-order fractional Fourier transform filter was used to filter each segment. Finally, the filtered signal was demodulated for gear failure diagnosis at an early stage. Romero et al. [94] used a machine learning algorithm to generate a baseline for identification of the normal operation conditions. Then the ICD method was used for gear fault type recognition in wind turbines. The abovementioned methods are summarized in Table 6.

### 3.2. Wavelet Transform Methods

A WT employs wavelets as the basis to extract the transient signals from the measured vibration signals. Fan et al. [95] proposed a wavelet-based statistical signal detection approach for monitoring and diagnosing the bearing compound faults at an early stage. He et al. [96] introduced a fuzzy rule to maximize the amplitudes of bearing characteristic frequencies (BCFs), then the sum of the BCFs amplitude and their harmonics were used for early fault detection in bearings. Cui et al. [97] utilized the wavelet transform, time–frequency analysis and blind source separation theory to determine the shared frequencies for early bearing fault diagnosis. Cui et al. [98] used WT to de-noise the signals, and then extracted the fault characteristics in both the time domain and the time-frequency domain. Finally, the fault location was identified using the grey correlation method. Wang et al. [99] proposed an adaptive wavelet stripping algorithm to extract fault information for early bearing fault detection. Combet et al. [100] used the instantaneous wavelet bicoherence as fault features for local damage detection in gears. Moumene et al. [101] used wavelets multiresolution analysis and the high-frequency resonance technique to demodulate signals for gear or bearing defects detection. Chen et al. [102] used the relative energy ratio to select the sensitive frequency bands of multiwavelet packet coefficients for compound faults detection of rotating machinery. He et al. [103] proposed maximal-overlap adaptive multiwavelet to extract the fault symptom for mechanical fault detection at an early stage. Li et al. [104] applied intrinsic characteristic-scale decomposition (ICD) to decompose the measured vibration signals into a set of PC components. Then the tunable Q-factor wavelet transform is used to generate the high and low Q-factor components. Finally, the envelope analysis is implemented to obtain the low Q-factor component for the EFD.

Discrete wavelet transform discretizes the scale and translation of basic wavelets. With other wavelet transforms, a key advantage over Fourier transform is temporal resolution: it captures both frequency and location information. Fan et al. [105] used discrete wavelet transform to reduce the noise and decompose the signal into several decomposition levels, and then statistical parameters were applied for detecting the gear damage. Rahman et al. [106] used a discrete wavelet transform to extract fault information, and then envelope detection was used for online condition monitoring of unbalanced rotor induction motor. Rangel-Magdaleno et al. [107] used a discrete wavelet transform to extract fault information for fault diagnosis of partially damaged rotor bars in induction motors under different load conditions. In [108], the wavelet coefficients which have been extracted over a wide range of signals and the analysis could be percolated on the frequency domain by HAAR wavelet and finally fault diagnosis of induction machines was realized.

The wavelet packet decomposition (WPD) is a wavelet transform, in which the signal is passed through more filters. Tse et al. [109] utilized envelope analysis to demodulate the wavelet packet coefficients which extracted from the selected wavelet packet node for the weak bearing fault signal detection. Yang et al. [110] developed a noise suppression method through combining EMD and WPD for weak feature extraction of wind turbine bearings.

The Morlet wavelet is a wavelet composed of a complex exponential Gaussian window (envelope). Morsy et al. [111] apply the optimal Morlet wavelet filter to preprocess the signal, and envelope detection was used to identify bearing failures at early stage. Yiakopoulos [112] proposed a feature extraction method combining the morphological analysis and complex shifted Morlet wavelets for bearing fault diagnosis. In [113], the signal was filtered using a band-pass filter determined by a Morlet wavelet which parameters were optimized based on maximum kurtosis. Then the envelope detection and the energy operator were used for gear fault diagnosis. Wang et al. [114] utilized a complex Morlet wavelet filter which is determined by a simplex-simulated annealing algorithm to extract the weak bearing fault features. The abovementioned methods are summarized in Table 7.

### 3.3. Sparse Representation Methods

Sparse representation aims to represent a signal accurately via a limited number of linear combinations of basic signals. Tang et al. [115] utilized sparse representation-based latent components decomposition to extract incomplete or seriously overwhelmed fault components for the EFD of bearings and gears. Lv et al. [116] used improved linear frequency-modulated function as an atom in atomic sparse decomposition for early weak fault detection. Mo et al. [117] proposed a delayed correlation envelope technique based on sparse signal decomposition method to extract early weak fault information. Cui et al. [118] used the sparse decomposition to analyze the early-stage bearing fault via adaptive impulse dictionary construction. Tang et al. [119] used the compressive sensing technique for dimensionality reduction, and then the sparse representation classification algorithm was applied for rotating machinery fault recognition. Li et al. [120] used resonance-based sparse signal decomposition (RSSD) to decompose the signal into high resonance component and low resonance component. Finally PCA was used to extract the principal components and transform the signals into frequency domain to locate the early fault of rolling bearing in motor. These methods are summarized in Table 8.

### 3.4. Other Fault Frequency Based Methods

Chaotic-based methods such as morphological operators, entropy, etc. have attracted increasing attention in early fault diagnosis. Aijun et al. [121] used morphological operators to extract impulsive signal for early fault diagnosis of rolling element bearings. Raj et al. [122] utilized a kurtosis-based algorithm to select structuring elements of morphological operators for early bearing fault detection. Dong et al. [123] combined the minimum entropy deconvolution and K-singular value decomposition for incipient bearing fault detection. Kedadouche et al. [124] used nonlinear parameters including approximate entropy, sample entropy, and Lempel-Ziv complexity as the extracted fault features for early detection of gear faults. Wang et al. [125] combined the minimum entropy de-convolution and fast kurtogram to extract early rolling bearing fault features.

Spectral kurtosis is very effective at processing the transient components embedded in weak fault signals. Many researchers have applied spectral kurtosis in the EFD of rotating machinery. Antoni [126] redefined the theory of spectral kurtosis in detail, its complete algorithm and its applications, and it is now widely used in the field of mechanical performance degradation. Based on the study of spectral kurtosis theory, Antoni [127] defined a new meaning of spectral kurtosis and applied it to detect early mechanical performance degradation. Meanwhile, a fast algorithm was proposed for computing the spectral correlation by Antoni et al. [128] for the early fault diagnosis of rolling element bearings. Cong et al. [129] combined spectral kurtosis with an autoregressive model to detect incipient fault of bearings. Jeong et al. [130] used spectral kurtosis to find the most informative sub-band signals for representing abnormal symptoms in bearing failures. Jia et al. [131] used maximum correlated kurtosis deconvolution (MCKD) to highlight the periodic impulse components, then the spectral kurtosis is applied to select the resonant frequency band of the signal filtered by MCKD and generate the envelop spectrum for diagnosing the bearing faults.

Li et al. [132] utilized the particle filter to reduce the noise. Then, the optimal band is chosen by using fast kurtogram. In the end, the fault character frequency can be obtained by using spectrum analysis. Chen et al. [133] used mean envelope kurtosis to determine the optimum frequency band, and envelope analysis was used to detect the bearing faults. Masmoudi et al. [134] proposed a reduced frequency model to provide an accurate estimation of the fault frequency. Dong et al. [135] proposed the frequency-shifted bispectrum to analyze amplitude modulated and frequency modulated signals for qualitative and quantitative diagnosis of bearing faults. Zhou et al. [136] proposed the cyclic bispectrum for early rolling element bearing fault diagnosis via bispectrum and cyclostationarity analysis. Based on cyclic spectral density, Dong et al. [137] proposed a Wigner–Ville spectrum which is a noise resistant time frequency analysis technique for extracting bearing fault patterns. Yuan et al. [138] used multi-fractal analysis in the time-frequency domain to identify fault types at an early stage. Siegel et al. [139] proposed a tachometer-less synchronously averaged envelope feature extraction technique for rolling element bearing health assessment. Park et al. [140] proposed the minimum variance cepstrum for early fault detection of bearings by filtering a logarithmic power spectrum. 

Fu et al. [141] put statistic features into adaptive fuzzy-means clustering to recognize different fault types of rolling bearings. Li et al. [142] proposed a fuzzy technique to select the informative frequency band for weak fault detection. Liu et al. [143] proposed adaptive SR to detect bearing faults based on the quantum particle swarm algorithm. Liao et al. [144] present an automatic filtering method based on improved genetic algorithm to extract fault signals and reduce the noise for fault type identification of rolling bearings. 

Javorskyj et al. [145] proposed the covariance and spectral characteristics of periodically correlated random processes to describe the state of rotary mechanical systems. Igba et al. [146] used RMS and peak values of vibration signals for condition monitoring of wind turbine gearboxes. Shao et al. [147] defined a new concept of RMS value calculation using angle domain signals within small angular ranges. Using the time pulses of an encoder, this new RMS could be used to diagnose helical gears fault at low rotational speeds. Sharma et al. [148] proposed modified time synchronous averaging to increase SNR, and statistical features were used to detect gear crack at different fluctuating speed. Jin et al. [149] developed Mahalanobis distance to indicate the health condition of cooling fan and induction motor. All the abovementioned methods are summarized in Table 9.

## 4. Applications of AI in Early Fault Diagnosis of Rotating Machinery

### 4.1. KNN

In a KNN-based fault diagnosis algorithm, the signal is first processed by other techniques to extract the fault features, aiming to constructing the fault feature vector. Then the fault feature vector is taken as the input of KNN to identify early fault of rotating machinery. Georgoulas et al. [150] transformed the raw signal into a discrete component, which was represented by a histogram form summarizing the occurrence of the chosen symbols. The histogram was taken as the input of KNN for fault location and severity identification of rolling bearing. Gao et al. [151] combined S transform with morphological pattern spectrum for feature extraction, and then KNN was used to identify bearing faults. Rajeswari et al. [152] used EEMD to extract features. Then a hybrid binary bat algorithm combined with a machine learning algorithm was used to reduce the dimensionality and select the predominant features. Finally, KNN was used to identify gear faults at an early stage. Geramifard et al. [153] proposed a squeezing and stretching method to preprocess the signal, and then a hidden Markov model was built. The parameters of this model were used as input of KNN for fault detection and diagnosis in synchronous motors. Holguín-Londoño [154] proposed filter bank methods to decompose bandwidth-limited signals into a set of narrow-band components. The similarity between the input signal and each extracted narrow-band component was used as fault features and KNN was used to identify different types of faults. The above mentioned methods are summarized in Table 10.

### 4.2. SVM

The SVM-based fault diagnosis is very similar to KNN-based methods. Fan et al. [155] used statistical parameters and characteristic amplitude ratios of frequency bands as fault features, then the PCA was used to reduce the fault feature dimensions. The selected features were put into SVM for gear fault type identification. Liu et al. [156] used the impact time frequency dictionary as fault features, and then SVM was trained for fault recognition. Fernández-Francos et al. [157] used band-pass filters and Hilbert transform to extract fault features and one-class SVM was used to discriminate between normal and faulty conditions. Shen et al. [158] used statistical feature as input, and a two-layer structure consisted of support vector regression machine was used to recognize bearing fault patterns and track the fault sizes. Saidi et al. [159] proposed a method for wind turbine high-speed shaft bearings which used spectral kurtosis as fault features, and support vector regression for fault classification.

Many researchers have applied SVM with adaptive decomposition techniques for bearing fault diagnosis in recent years. Zhao et al. [160] used EEMD to decompose the signal into IMFs, and multi-scale fuzzy entropy value of selected IMF was used to form the fault features. The SVM was used to diagnose the fault types and fault severities for motor bearing. Fan et al. [161] used EMD to decompose the signal into IMFs, then the characteristic energy ratios of IMFs and other statistical parameters were used as input of SVM for gear tooth surface damage diagnosis. Tabrizi et al. [162] used WPD to denoise the signal, the fault features extracted by EEMD were put into SVM to detect small defects on roller bearings under different operating conditions. Wu et al. [163] used CWT to decompose the signal, then put wavelet coefficients into a tree kernel-based SVM to identify bearing faults. Konar et al. [164] used CWT and Hilbert transforms to extract fault information, then rough set theory (RST) and a genetic algorithm (GA) were used to refine the fault features. Finally, SVM was used to detect six types of induction motor faults. Kang et al. [165] applied a singular value decomposition (SVD) algorithm for feature extraction, and a multi-layer support vector machine was used for fault classification of induction motors. The above mentioned methods are summarized in Table 11.

### 4.3. Neural Network

Neural networks can be used as classifiers in fault diagnosis, which is similar as the KNN and SVM methods. Jedlinski et al. [166] used CWT to extract fault features and a multilayer perceptron network was used for gear fault recognition. Bin et al. [167] extracted fault features by WPT and EMD, and then a BP neural network was used to identify early cracks of rotors. Soleimani et al. [168] used chaotic behavior features as fault features including the largest Lyapunov exponent, approximate entropy and correlation dimension, and then a neural network was used to identify different faults in rotating machinery. 

Several researchers have proposed improved neural networks methods, which can use the raw signals as the input without preprocessing. Eren et al. [169] put the raw vibration signals into a 1D convolutional neural network for a fast and accurate bearing fault detection. Chen et al. [170] used the raw vibration signals as the input of a multi-layer neural network for gearbox fault diagnosis. The above mentioned methods are summarized in Table 12.

### 4.4. Other Methods

This section describes other AI-based early fault diagnosis methods. Martin-del-Campo et al. [171] used dictionary learning to automatically derive signal features for characterizing different operational conditions and faults of rolling bearings. Almeida et al. [172] used time-domain features as input of generic multi-layer perceptron for bearing fault identification. Li et al. [173] used the non-dimensional symptom parameters after wavelet transformation to extract fault features, and then colony optimization was used to classify fault types at an early stage. Brkovic et al. [174] used the wavelet transform to decompose the vibration signals, then the standard deviation of average energy and the logarithmic energy entropy were used as the fault features. Finally, the quadratic classifier was used for early fault detection and diagnosis in bearings. Li et al. [175] used statistical parameters in both the time domain and the frequency domain to extract the fault features. Then the features are put into fuzzy lattice neurocomputing classification model to detect gear faults in an early stage. Cruz-Vega et al. [176] used discrete wavelet analysis to extract features, and then binary classification tree was used to identify different damage levels of rotor bars under different load conditions. Martínez-Rego et al. [177] put time domain features into one-class classifier for the early fault detection of rotating machinery. The above mentioned methods are summarized in Table 13.

## 5. Discussion and Conclusions

Based on the above descriptions, we can find that early fault diagnosis (EFD) of rotating machinery has achieved a large number of successful applications. Since the literature on this subject is huge and diverse, a review of all of the literature is impossible, and omission of some papers would be inevitable. From the previous section, some issues that should be studied in-depth for the real EFD applications are pointed out as follows:(1)Research on EFD based on multi-information fusion should be developed. In real applications, usually, multiple channel signals are measured simultaneously, such as vibration signals, current signals, torque signals, and rotating encoder signals. The extension of EFD techniques to multivariate versions can extract more characteristic fault information, which is vital for detection of weak fault symptoms at an early fault stage.(2)The calculation efficiency of EFD techniques deserves further research. Many EFD methods are proposed to improve the early fault detection ability at the cost of time consumption, which cannot meet the requirements of online condition monitoring. Therefore, how to improve the calculation efficiency of EFD is another research topic for early fault detection.(3)Most EFD methods are tested to be powerful on one test rig and the reliability test results on other machines are unknown. In real applications, the robustness of EFD methods should be studied, aiming to be effective for multiple machines.

## 6. Summary

Early fault diagnosis (EFD) of rotating machinery is essential to reduce the incidence of catastrophic failures and heavy economic losses. A review on the EFD of rotating machinery is presented in this paper. In this review, a number of EFD techniques for rotating machinery are surveyed in terms of the FFD-based early fault diagnosis and AI-based early fault diagnosis. In the end, some new research prospects are pointed out. It is believed that this review has synthesized the state-of-art references on EFD of rotating machinery for readers interested in this research area. Meanwhile, with the development of signal processing techniques, research on EFD of rotating machinery is expected to continue.

## Figures and Tables

**Figure 1 entropy-21-00409-f001:**
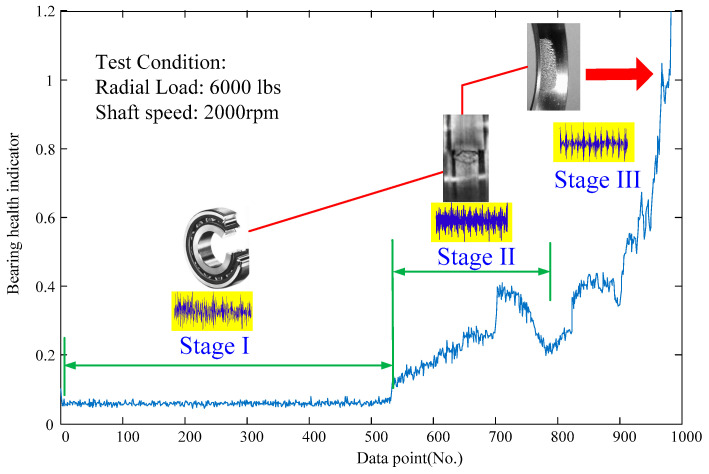
Amplitude of the anomaly measure versus the time point for a real bearing of whole life.

**Figure 2 entropy-21-00409-f002:**
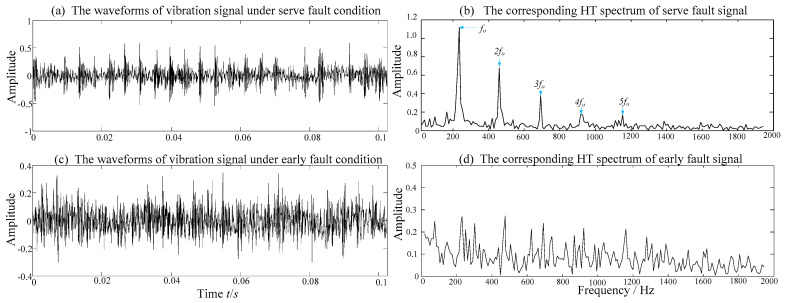
Two vibration signals under serve and early fault conditions and their corresponding Hilbert transform spectra: (**a**) the waveforms of vibration signal under serve fault condition, (**b**) the envelope spectrum of serve fault signal, (**c**) the waveform of vibration signal under early fault condition, (**d**) the envelope spectrum of early fault signal.

**Figure 3 entropy-21-00409-f003:**
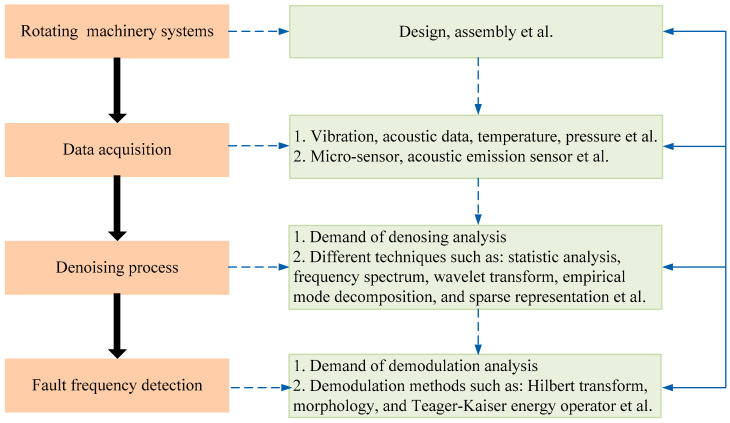
Early fault diagnosis framework using fault frequency detection method for early fault diagnosis of rotating machinery systems.

**Figure 4 entropy-21-00409-f004:**
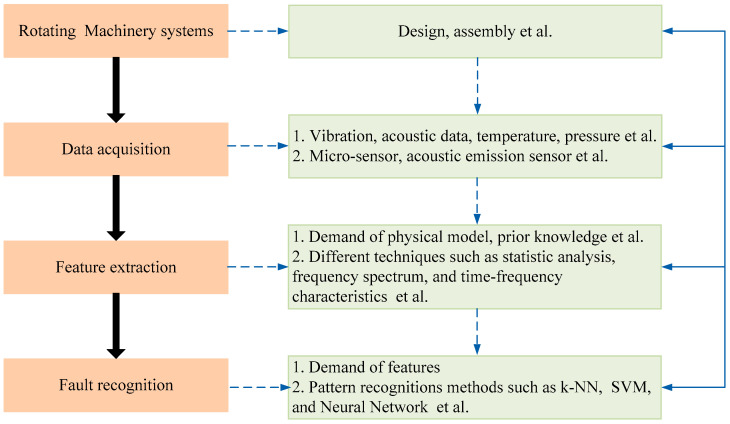
Early fault diagnosis framework using AI techniques for rotating machinery systems.

**Figure 5 entropy-21-00409-f005:**
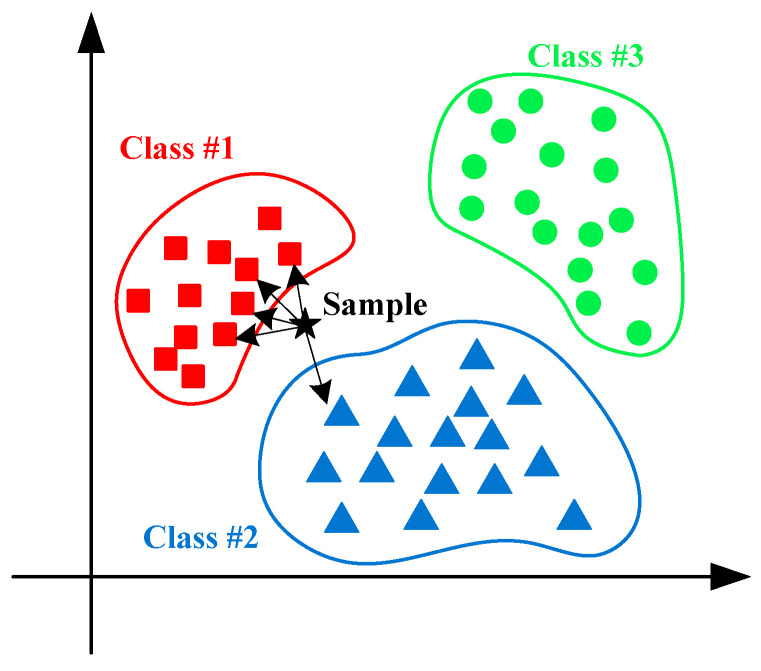
Diagram of the KNN method.

**Figure 6 entropy-21-00409-f006:**
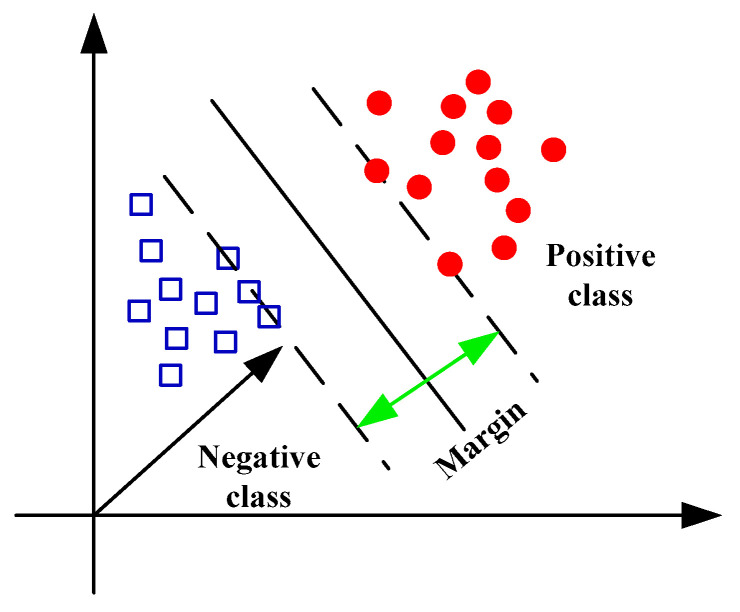
The optimal hyperplane for a binary classification by SVM.

**Figure 7 entropy-21-00409-f007:**
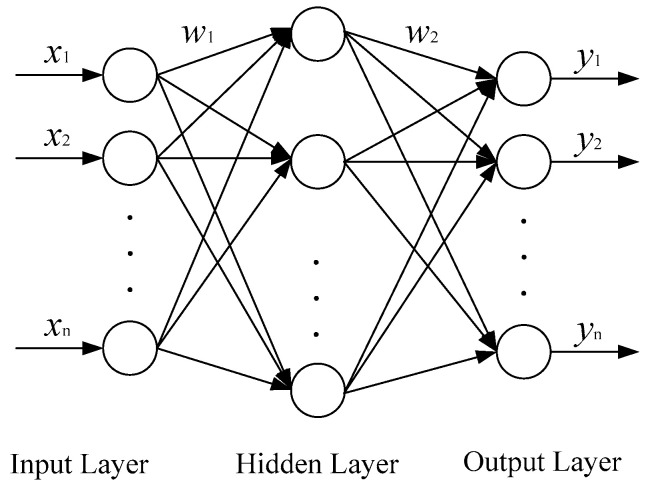
The structure of a BP neural network.

**Table 1 entropy-21-00409-t001:** Applications of empirical mode decomposition (EMD) method in early fault diagnosis of rotating machinery.

Authors	Methodologies
Dybała et al. [60]	EMD
Zhu et al. [61]	EMD + correlation coefficient
Dybała et al. [62]	EMD
Li et al. [63]	Bandwidth EMD + adaptive multiscale morphological analysis
Zhao et al. [64]	Approximate entropy + EMD
Lv et al. [65]	Multivariate EMD
Parey et al. [66]	EMD + variable cosine window

**Table 2 entropy-21-00409-t002:** Applications of ensemble empirical mode decomposition (EEMD) algorithm in early fault diagnosis of rotating machinery.

Authors	Methodologies
Guo et al. [71]	EEMD + similarity criterion
Imaouchen et al. [72]	Complementary EEMD
Li et al. [73]	Complementary EEMD
Tabrizi et al. [74]	Performance improved EEMD
Wang et al. [68]	EEMD + tunable Q-factor wavelet transform
Žvokelj et al. [69]	Independent component analysis multivariate monitoring + EEMD
Chen et al. [67]	EEMD + Hilbert Square Demodulation
Chen et al. [70]	EEMD + adaptive stochastic resonance
Jiang et al. [75]	EEMD + multiwavelet packet

**Table 3 entropy-21-00409-t003:** Applications of local mean decomposition (LMD) method in early fault diagnosis of rotating machinery.

Authors	Methodologies
Li et al. [79]	Differential rational spline-based LMD
Liu et al. [76]	LMD
Feng et al. [77]	LMD
Wang et al. [78]	LMD

**Table 4 entropy-21-00409-t004:** Applications of empirical wavelet transform (EWT) method in early fault diagnosis of rotating machinery.

Authors	Methodologies
Chen et al. [81]	Wavelet spatial neighboring coefficient + EWT
Boualem et al. [82]	EWT + Hilbert Transform
Zhang et al. [80]	Bistable stochastic resonance + EWT
Lu et al. [83]	Kurtogram + EWT + sparse regression

**Table 5 entropy-21-00409-t005:** Applications of variational mode decomposition (VMD) method in early fault diagnosis of rotating machinery.

Authors	Methodologies
Ma et al. [84]	Adaptive scale space spectrum segmentation + VMD + Teager energy operator
Li et al. [85]	Improved autoregressive-Minimum entropy deconvolution + VMD
Yang et al. [86]	Optimized VMD + simulated annealing
Guo et al. [87]	VMD + parameter optimization
Han et al. [88]	Rescaling subsampling compression + analytical mode decomposition + VMD
Jiang et al. [89]	EMD + VMD

**Table 6 entropy-21-00409-t006:** Applications of other adaptive method in early fault diagnosis of rotating machinery.

Authors	Methodologies
Elasha et al. [90]	Least mean squares (LMS)+fast block LMS
Zhao et al. [91]	Reweighted singular value decomposition
Ibrahim et al. [92]	Least mean squares algorithm
Mei et al. [93]	Multi-order self-adaptive filter
Romero et al. [94]	Machine learning + intrinsic characteristic-scale decomposition

**Table 7 entropy-21-00409-t007:** Applications of wavelet transform method in early fault diagnosis of rotating machinery.

Authors	Methodologies
Fan et al. [95]	Wavelet transform
He et al. [96]	Wavelet transform
Cui et al. [97]	Wavelet transform + time–frequency analysis + blind source Separation theory
Morsy et al. [111]	Morlet wavelet Filter + envelope detection
Yiakopoulos [112]	Morphological + Complex Shifted Morlet Wavelets.
Cui et al. [98]	High-frequency characteristics + self-adaptive wavelet de-noising
Wang et al. [114]	Complex Morlet wavelet coefficients + sparsity measurement
Tse et al. [109]	Wavelet transform + envelope analysis
Wang et al. [99]	Adaptive wavelet stripping algorithm
Morsy et al. [113]	Maximum Kurtosis + Morlet wavelet
Combet et al. [100]	Wavelet bicoherence
Moumene et al. [101]	Wavelets multiresolution analysis + the high-frequency resonance
Fan et al. [105]	Discrete wavelet transform
Karuppaiah et al. [108]	HAAR wavelet
Rahman et al. [106]	Discrete wavelet transform
Rangel-Magdaleno et al. [107]	Discrete wavelet transform + motor current signature analysis
Chen et al. [102]	Adaptive redundant multiwavelet packet
He et al. [103]	Adaptive multiwavelet
Yang et al. [110]	EMD + autocorrelation de-noising + wavelet package decomposition
Li et al. [104]	Intrinsic character-scale decomposition + tunable Q-factor wavelet transform.

**Table 8 entropy-21-00409-t008:** Applications of sparse decomposition method in early fault diagnosis of rotating machinery.

Authors	Methodologies
Lv et al. [116]	Atomic sparse decomposition + genetic algorithm
Li et al [120]	Resonance-based sparse signal decomposition + principal component analysis
Tang et al. [115]	Shift-invariant sparse coding
Mo et al. [117]	Delayed correlation envelope+ sparse decomposition
Cui et al. [118]	Sparse decomposition + adaptive impulse dictionary
Tang et al. [119]	Sparse representation + compressive sensing

**Table 9 entropy-21-00409-t009:** Applications of other fault frequency based method in early fault diagnosis of rotating machinery.

Authors	Methodologies
Aijun et al. [121]	Morphological operators
Raj et al. [122]	Morphological operators + fuzzy system
Dong et al. [123]	Minimum entropy deconvolution + K-singular value decomposition
Antoni J. [126]	Short-time Fourier-transform-based estimator of the spectral kurtosis
Antoni J. [127]	Fast computation of the kurtogram
Li et al. [132]	Particle Filter + Kurtogram
Wang et al. [125]	Minimum entropy de-convolution + Fast Kurtogram
Cong et al. [129]	Spectral kurtosis + autoregressive model
Jeong et al. [130]	Spectral kurtosis
Chen et al. [133]	Mean envelope Kurtosis + envelope analysis
Jia et al. [131]	Maximum correlated kurtosis deconvolution
Masmoudi et al. [134]	Time synchronous averaging
Dong et al. [135]	Frequency-shifted bispectrum
Zhou et al. [136]	Cyclic bispectrum
Dong et al. [137]	Wigner–Ville spectrum
Yuan et al. [138]	Multi-fractal analysis
Siegel et al. [139]	Tachometer-less synchronously averaged envelope
Park et al. [140]	Minimum variance cepstrum
Fu et al. [141]	Adaptive fuzzy-means clustering
Li et al. [142]	Informative frequency band
Liu et al. [143]	Adaptive SR + quantum particle swarm
Liao et al. [144]	Improved genetic algorithm
Kedadouche et al. [124]	Approximate entropy + sample entropy + Lempel-Ziv Complexity.
Javorskyj et al. [145]	Periodically correlated random processes
Igba et al. [146]	Root mean square (RMS) + peak values
Shao et al. [147]	RMS in angle domain
Sharma et al. [148]	Modified time synchronous averaging
Jin et al. [149]	Mahalanobis distance

**Table 10 entropy-21-00409-t010:** Applications of k nearest neighbor (KNN) method in early fault diagnosis of rotating machinery.

Authors	Methodologies
Georgoulas et al. [150]	Symbolic Aggregate approximation + KNN
Gao et al. [151]	Stransform + morphological pattern spectrum + KNN
Rajeswari et al. [152]	EEMD + hybrid binary bat + KNN
Geramifard et al. [153]	Hidden Markov model + KNN
Holguín-Londoño [154]	Filter bank + KNN

**Table 11 entropy-21-00409-t011:** Applications of support vector machine (SVM) method in early fault diagnosis of rotating machinery.

Authors	Methodologies
Shen et al. [158]	Statistical feature + SVM
Liu et al. [156]	Impact time frequency dictionary + SVM
Fernández-Francos et al. [157]	Band-pass filters and Hilbert Transform + ν-SVM
Zhao et al. [160]	EEMD + multi-scale fuzzy entropy + SVM
Tabrizi et al. [162]	WPD + EEMD + SVM
Wu et al. [163]	Continuous wavelet transform+ SVM
Fan et al. [155]	Statistical parameters + PCA + SVM
Kang et al. [165]	Singular value decomposition+ SVM
Konar et al. [164]	CWT + GA + SVM
Saidi et al. [159]	Spectral kurtosis + SVM

**Table 12 entropy-21-00409-t012:** Applications of neural network method in early fault diagnosis of rotating machinery.

Authors	Methodologies
Eren et al. [169]	1D convolutional neural networks
Jedlinski et al. [166]	CWT + multilayer perceptron network
Chen et al. [170]	Multi-layer neural networks
Bin et al. [167]	Wavelet packet transform+ EMD + BP neural network
Soleimani et al. [168]	Chaotic behavior features + neural network

**Table 13 entropy-21-00409-t013:** Applications of other AI-based method in early fault diagnosis of rotating machinery.

Authors	Methodologies
Martin-del-Campo et al. [171]	Dictionary learning
Almeida et al. [172]	Time-domain features + generic multi-layer perceptron
Li et al. [173]	Wavelet transformation + ant colony optimization
Brkovic et al. [174]	Wavelet transformation + quadratic classifier
Li et al. [175]	Fuzzy lattice neurocomputing
Cruz-Vega et al. [176]	Discrete wavelet + binary classification tree
Martínez-Rego et al. [177]	Time domain features + one-class classifier

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
