# Peer review of "A Review of Early Fault Diagnosis Approaches and Their Applications in Rotating Machinery"

_entropy, 2019, doi:10.3390/e21040409_

Round 1

Reviewer 1 Report

Generally I recommend to publish the manuscript. It is well organized and of high scientific quality. Regarding to the problems described by authors I would suggest to refer to the problem of application of 2D wavelet transform to detect surface irregularities of machine parts, which was for example given in the work 

Stępień K., Makieła W.: An analysis of deviations of cylindrical surfaces with the use of wavelet transform , Metrology and Measurement Systems, Vol. XX (2013), No. 1, pp. 139−150.

Author Response

Response: Thanks for your comments. In the revised paper, we have cited this reference.

Reviewer 2 Report

1) It would be advised to include in such wide review of approaches authors known in the field. For instance, works of J. Antoni are completely ignored, while modifications of his works are cited.

2) Overall, review is well-written. One thing that can raise objections is minor presence of authors that contributed to this field, while authors that have written about modifications are cited.

Author Response

Response: Thanks for your comments. We have cited the works of J. Antoni in the revised paper. Meanwhile, we have added more important works in this field in this revised paper as follows.

[127] Antoni, J. The spectral kurtosis: an useful tool for characterizing non-stationary signals. Mechanical systems and signal processing, 2006, 20,282-307.

[128] Antoni, J. Fast computation of kurtogram for the detection of transient faults. Mechanical systems and signal processing, 2007, 21, 108-124.

[129] Antoni J., Xin G., Hamzaoui N. Fast computation of the spectral correlation. Mechanical systems and signal processing, 2017, 92, 248-277.

Reviewer 3 Report

The issue this manuscript trying to address is interesting and important. However, there critical concerns with the paper.

Extensive editing of English language and style is required

A few Examples:

-          “Timely” as adv is archaic . \ref{Cambridge online dictionary}

-          L13,14: “Massive… FED techniques”

-          L28: “safety opperation”, safe operation.

-          L36,37: “However,…impulses”

-          L60: “and difficult research task on engineering applications”.

Typo:

-          L18: methods not method

And many others.

Conceptual concerns

Minor conceptual concerns:

-          L9,23: “…. Modern industrial applications”

Rotatory machineries were among the first mechanical system ever used!

-          L27: “Weak faults!”

-          L25: “Due to the harsh working environment, ….”

System are prone to faults after they are in service (duty cycles), and harsh environmental conditions can trigger it!

-          L38: “strong noise!” please comment scientifically, like S/N ratio…, and high power noise is not only due to the transfer path!

-          Fig2/ L51:58: Noise power in both should be the same, it is a property of the sensor! Why different in a and b? The things which is changed is S/N ratio.

What is the vibration signal?

-          Coined terms like “Frequency fault detection” doesn’t convey the meaning of frequency-domain fault detection methods! Please use the stablished terminology in the field.

-          L73:85: All should be deleted!!

Major conceptual concerns:

-          In review paper, it is not common to present results without references like Fig.2.

Section2. Time domain methods are completely ignored while many practical applications. The classification of frequency based method and AI method are not aligned!

-          Time domain methods

-          Frequency domain methods

-          Time-Frequency domain methods

And AI can be used in any of them.

-          Wavelet transform is a Time-Frequency domain method. Not frequency domain method.

Author Response

Recommendation 1:

A few Examples:

(1) “Timely” as adv is archaic . \ref{Cambridge online dictionary}

Response: Thanks. We have deleted “Timely” in the revised paper.

(2) L13,14: “Massive… FED techniques”

Response: Thanks. We have replaced this expression in the revised paper as follows.

Massive research effort work has been conducted in last two decades to develop EFD techniques.

(3) L28: “safety opperation”, safe operation.

Response: Thanks. The revision is done exactly as you suggested.

(4) L36,37: “However,…impulses”

Response: Thanks. The revision is done exactly as you suggested.

(5) L60: “and difficult research task on engineering applications”.

Response: Thanks. We have deleted this sentence in the revised paper.

(6) L18: methods not method.

Response: Thanks. The revision is done exactly as you suggested.

Recommendation 2:

Minor conceptual concerns:

 (1) L9,23: “…. Modern industrial applications”

Rotatory machineries were among the first mechanical system ever used!

Response: Thanks for your comments. We have deleted “modern”in the revised paper. 

(2)  L27: “Weak faults!”

Response: Thanks for your comments. We revised the “weak fault” into “early fault” in the revised paper.

(3) L25: “Due to the harsh working environment, ….”

System are prone to faults after they are in service (duty cycles), and harsh environmental conditions can trigger it!

Response: Thanks for your comments. We revised this sentence in the revised paper as follows:

Due to high service load, harsh operating conditions or inevitable fatigue, faults may develop in rotating machinery.

Detailed revisions are available in L25 of page 1.

(4) L38: “strong noise!” please comment scientifically, like S/N ratio…, and high power noise is not only due to the transfer path!

Response: Thanks for your comments. We have deleted the expression “the long transfer path” in the revised paper.

(5) Fig2/ L51:58: Noise power in both should be the same, it is a property of the sensor! Why different in a and b? The things which is changed is S/N ratio.

Response: Thanks for your comments. From your comment, we find that there is something wrong in the explanation of Fig.2. We have revised this error. Meanwhile, we have added one more reference in Fig.2 for better understanding. The detailed descriptions are given as follows.

Two vibration signals taken from the severe fault stage (the phase III) and early fault stage (phase II) are shown in Fig.2 (a) and Fig.2 (c), respectively.

Detailed revisions are available in L51 of page 2.

 (6) Coined terms like “Frequency fault detection” doesn’t convey the meaning of frequency-domain fault detection methods! Please use the stablished terminology in the field.

Response: Thanks for your comments. The fault frequency detection (FFD) technique includes both the frequency-domain methods and time-frequency domain methods. This study utilizes this expression to summarize the research work of fault frequency-based methods.

 (7)  L73:85: All should be deleted!!

Response: Thanks for your comments. We have deleted these sentences in the revised paper.

Recommendation 3:

Major conceptual concerns:

(1) In review paper, it is not common to present results without references like Fig.2.

Response: Thanks for your comments. In the revised paper, we have added the corresponding reference in Fig.1 and Fig.2.

[31] Li Y.; Xu M.; Wei Y.; Huang W Health condition monitoring and early fault diagnosis of bearings using SDF and intrinsic characteristic-scale decomposition. IEEE Transactions on Instrumentation and Measurement, 2016, 9: 2174-2189.

(2) Section2. Time domain methods are completely ignored while many practical applications. The classification of frequency based method and AI method are not aligned!

Time domain methods

Frequency domain methods

Time-Frequency domain methods

And AI can be used in any of them.

Response: Thanks for your comments. All the time domain methods, frequency domain methods and time-frequency domain methods are included in this review study. The structure is suitable to summarize the research work in the field of early fault diagnosis (EFD). Details are explained as follows.

First, this study reviews the applications of EFD of rotating machine are reviewed in two aspects: fault frequency-based methods and artificial intelligence-based methods. The fault frequency detection (FFD) technique includes both the frequency-domain methods and time-frequency domain methods. The artificial intelligence-based methods can cover time domain methods, frequency domain methods and time-frequency domain methods.  

Second, the review study is directly based on the final diagnostic form. This structure will facilitate other researchers especially new researchers to properly select appreciate EFD method to diagnose the early fault. Meanwhile, our work will help them and save their time to find proper methods for their specific applications.

(3) Wavelet transform is a Time-Frequency domain method. Not frequency domain method.

Response: Thanks for your comments. Yes, you are right. Wavelet transform is a time-frequency domain method. However, the fault frequency detection (FFD) technique includes both the frequency-domain methods and time-frequency domain methods. Therefore, the early fault diagnosis using wavelet transform is covered in Section 2.

 Details see the response of Recommendation 2 (6). 
